# A Structural Model of Quality of Life in Patients after Colorectal Cancer Surgery

**DOI:** 10.3390/ijerph19052564

**Published:** 2022-02-23

**Authors:** Jeong Won Yeom, Yeon Ok Suh

**Affiliations:** 1Department of Nursing, Soonchunhyang University Bucheon Hospital, 170, Jomaru-ro, Bucheon-si 14584, Korea; 2School of Nursing, Soonchunhyang University, 31 Soonchunhyang 6th Rd, Dongnam-gu, Cheonan 31151, Korea; yeonok@sch.ac.kr

**Keywords:** colorectal cancer, quality of life, self-efficiency, health promotion behavior, structural model

## Abstract

Background: This study was conducted to determine a hypothetical model for the factors affecting the quality of life of postoperative colorectal cancer patients. Methods: We collected data from 209 patients that was analyzed using SPSS 22.0 and AMOS 25. Results: Predictive variables of the hypothesis model included an exogenous variable, social support, as well as endogenous variables self-efficacy, symptoms, health promotion behavior, and quality of life. Testing of the correction model showed that significant paths directly affecting quality of life of patients after colorectal cancer surgery included health promotion behavior, symptoms, and self-efficacy and also showed an explanation power of 58.7%. Social support was found to have a significant impact on the quality of life indirectly through self-efficiency. Conclusions: To improve the quality of life of colorectal cancer patients, it is necessary to develop a nursing intervention program that strengthens patients’ health promotion behaviors to alleviate their symptoms and improve their social support and self-efficacy.

## 1. Introduction

Colorectal cancer is the second most common cancer among men and women in Korea as lifestyles become more westernized, and the incidence rate is increasing every year.

Also the 5-year survival rate increased to 71.8% [1]. The principle of treatment for colorectal cancer is to achieve a histologically disease-free state through radical resection, and after surgery, adjuvant treatment with chemotherapy is used for high-risk stage 2 or higher depending on the histological stage. In the case of rectal cancer, chemotherapy and radiotherapy are administered before or after surgery as adjuvant treatment to reduce recurrence in the pelvis [2]. Unlike in the past, when cancer diagnosis meant death, due to the development of early diagnosis of cancer through national cancer screening projects and auxiliary treatment such as surgery, chemotherapy, and radiation therapy, it is now recognized as a chronic disease that can be survived for a long time [3]. Therefore, in addition to increasing the survival rate of cancer patients, the final treatment goal for colorectal cancer patients is to improve the quality of life of the subjects [2].

Quality of life means a subjective sense of well-being in one’s life [4]. In particular, in order to evaluate and improve the quality of life of cancer patients, it has been evaluated including various factors such as physical, psychological, social well-being, and symptoms, as well as the social context in which the patient belongs [5]. Recently, the importance of quality of life as an indicator or predictor of survival of cancer patients has been emphasized [6], and the improvement of health promotion behaviors and quality of life of the subject is being evaluated in relation to it [7].

Since research using Wilson and Cleary’s health-related quality of life model was introduced as a research model to understand quality of life, many domestic and foreign studies have been reported applying the main concept of this model. In this study, a theoretical framework of factors influencing the quality of life of colon cancer surgery patients was also created. In studies conducted on elderly patients [8], coronary artery disease [9], and thyroid cancer [10], the main concepts proposed by this model are physiological characteristics that have been reported on, including symptom status, functional status, disease awareness, environmental characteristics, and quality of life. This model was applied to clarify the quality of life elements of various topics by reporting the relationship between these main concepts and quality of life.

As for factors affecting the quality of life of colorectal cancer patients, social support [11], symptom experience [12], self-efficacy [13], and health promotion behavior [14] were reported as major variables in Wilson and Cleary’s model [4]. As factors highly related to the proposed main concepts, no studies have applied this model to colorectal cancer patients in Korea. In this country, the number of colorectal cancer patients has recently increased, and the 5-year survival rate has increased compared to other cancers [1]. Since colorectal cancer patients should be interested in their quality of life after cancer diagnosis, it is necessary to analyze the factors that directly or indirectly affect their quality of life.

Looking at the characteristics constituting the research model, environmental characteristics have been studied as social support factors, and several studies have shown that they have a consistent effect on quality of life [9,11]. The decrease in social support after colorectal cancer surgery is related to the increased burden on family members due to the disease [11]. In particular, less than 12 months after surgery, social support was an important factor influencing the quality of life [15]. In addition, patients with low social support showed lower quality of life in all physical, functional, cognitive, and social domains than patients with high social support [16]. It has been consistently reported that social support is a very important factor in improving the quality of life of cancer patients. 

Symptom experience characteristic are related to the fact that colorectal cancer patients experience more digestive symptoms than other types of cancer. In particular, it can be seen that symptoms related to defecation (e.g., feeling urgency, frequent defecation, and fecal incontinence), dry mouth, and changes in taste after surgery reduce quality of life [12]. In addition, psychological pressure and physical and mental discomfort arising from the treatment process [17] can greatly affect quality of life because patients experience physical symptoms related to digestive characteristics [18].

Studies have been reported that suggest self-efficacy as an individual characteristic in the quality of life model [8]. For colorectal cancer patients, self-efficacy to manage their symptoms or disease-related problems was a very important factor even after diagnosis and during recovery, and this was found to affect quality of life [19]. Subjects with high self-efficacy managed pain and symptoms better than subjects with low self-efficacy, and had a higher psychological well-being, resulting in higher health-related quality of life [20].

The health promotion behavior of cancer patients is important for the prevention of complications that occur during the treatment process. In particular, intervention studies on lifestyle changes to prevent recurrence of cancer have been reported. In this study, interventions for changes in healthy lifestyles are short-term but helpful in improving health [21], and proper physical activity, dietary fiber intake, alcohol reduction, smoking cessation, and maintenance of appropriate weight for colorectal cancer patients were suggested. It can be seen that these health promotion behaviors are important factors in preventing recurrence of colorectal cancer, increasing health promotion and survival rates [22], and that health promotion behaviors are powerful factors in improving the quality of life of cancer patients [23]. 

Social support and self-efficacy of cancer patients are major factors that increase health promotion behaviors [24], and social support plays a role as a mediator to successfully lead cancer patients’ life in battling the disease and maintaining health promotion behaviors [25]. In addition, self-efficacy to manage symptoms is as important as measuring the symptom level of colorectal cancer patients [19], suggesting that self-efficacy is a major factor in changing health promoting behaviors of cancer patients [26]. 

The characteristics of subjects related to the quality of life of colorectal cancer patients were reported as significant influencing factors, including gender, age, occupational status, spouse status, and economic status [27,28]. Among the factors suggested by Wilson and Cleary [4], factors influencing the quality of life of colorectal cancer surgery patients were selected as relating to self-efficacy, and environmental characteristic variables were selected as relating to social support. Symptom state variables were selected as related to symptom experience, and functional state variables were selected as related to health promotion behavior, and a hypothetical model of this study was established. Also the individual characteristics to understand the factors affecting the quality of life of colorectal cancer surgery patients were selected as self-efficacy, and the environmental characteristics were selected as social support. A hypothetical model was established with the symptom state as symptom experience and the functional state as health promotion behavior. In this hypothetical model, exogenous variables are social support, and endogenous variables are self-efficacy, symptom experience, health promotion behavior, and quality of life. Measured variables for exogenous variables are medical support, family support, and friend support of colorectal cancer surgery patients. Measured variables for endogenous variables are general symptoms and digestive symptoms, which are sub-factors of symptom experience, and sub-factors of health promotion behavior are: self-actualization, health responsibility, exercise, nutrition, interpersonal relationships, rest and stress. There are four sub-factors of quality of life: physical state, social/family state, emotional state, and functional state. Therefore, in this study, a model was constructed with 16 measurement variables for one endogenous variable and four exogenous variables (Figure 1).

Thus based on the health-related quality of life model of Wilson and Cleary [4], this study reflects the personal characteristics of self-efficacy and environmental factors identified as factors affecting the quality of life of patients with colorectal cancer surgery in previous studies. The purpose of this study is to establish a model and verify the effect on the quality of life of patients after colorectal cancer surgery.

### Research Aims

This study aims to investigate the factors affecting the quality of life of patients after colorectal cancer surgery based on the health-related quality of life model of Wilson and Cleary [4]. 

## 2. Materials and Methods

### 2.1. Research Design

This study is based on the health-related quality of life model of Wilson and Cleary [4] and previous studies [9,10]. This study established a hypothetical model by synthesizing the influencing factors that explain the quality of life of patients with colorectal cancer surgery.

This is a structural model validation study that tests the proposed hypothesis. First presents a virtual model for the quality of life of colon cancer surgery patients. Second, a modified model is presented that verifies the suitability between the virtual model and actual data and better explains the relationship between related variables. Third, the direct and indirect effects of variables affecting the quality of life of colorectal cancer patients are confirmed.

### 2.2. Criteria for Selection 

The subjects of this study were convenience extractions from outpatients who visited Soonchunhyang University Hospital in Gyeonggi-do for follow-up management after colorectal cancer surgery. A sample size greater than 200 was required for structural equation model analysis [29]. Subject selection criteria are patients who have completed treatment such as chemotherapy or radiation therapy after colon surgery, have normal cognitive function, are able to communicate, understand the purpose of the study, and agree in writing to participate.

### 2.3. Study Design and Sample

A cross-sectional survey design was employed, and 209 participants with type 2 diabetes were recruited from outpatient clinics at university hospitals in South Korea from November 2018 to May 2019. 

### 2.4. Instruments and Measures

#### 2.4.1. Social Support

Social support was measured using the Multidimensional Scale of Perceived Social Support (MSPSS), which consists of 12 items organized into three sub-domains (family, friend, and significant neighbor) [30]. Responses used a five-point Likert scale (1 = not at all, 5 = very strongly agree). Higher scores indicate greater support from family, friends, and acquaintances. This scale’s Cronbach’s was 0.85 in the originating research and 0.91 in the present research. 

#### 2.4.2. Self-Efficacy 

Self-efficacy was assessed using the Self Efficacy Scale, which consists of 13 items and uses a five-point Likert scale (1 = very strongly disagree, 5 = very strongly agree) [31]. Higher scores indicated greater self-efficacy. This scale’s Cronbach’s was 0.83 in the originating research and 0.89 inter present research.

#### 2.4.3. Symptom Experience

Symptom experience was measured using the MD Anderson Symptom Inventory—Gastrointestinal Cancer Module (MDASI-GI), which consists of 13 symptom questions commonly experienced by cancer patients such as pain and fatigue and five digestive symptoms (constipation, diarrhea, dyspnea, changes in taste, and bloating). Each question is a measure of “no” (0 point) to “imaginably severe” (10 points), and the range of scores ranges from 0 to 180 points, meaning that the higher the score, the higher the degree of symptom experience. This scale’s Cronbach’s was 0.80 in the originating research and 0.95 inter present research.

#### 2.4.4. Health Promoting Life Style Profile

Health promoting lifestyle was measured using the Health Promoting Lifestyle Profile (HPLP) [32]. The HPLPS is composed of sub-scales such as physical activity, nutrition, moral development, relationship between individuals, and stress management behaviors. In the assessment, Likert scale was used for the analysis as follows: never = 1, sometimes = 2, frequently = 3, regularly = 4. This scale’s Cronbach’s was 0.92 in the originating research and 0.94 inter present research.

#### 2.4.5. Quality of Life 

Quality of life was measured using the Functional Assessment of Cancer Therapy–Colon (FACT-C) questionnaire [33]. The FACT-C consists of 36 items. Participants rated these items on a five-point scale for each of the following subscales: physical (7 items), social/family (7 items), emotional (6 items), functional well-being (7 items) and colon-specific symptoms (9 items). Each subscale score was calculated by taking the sum of the item scores and dividing by the number of items answered. Higher scores indicate a better QoL. This scale’s Cronbach’s a was 0.91 in the originating research and 0.89 inter present research.

### 2.5. Ethical Considerations

This study passed an IRB review from Soonchunhyang University Bucheon Hospital (IRB 2018-05-012-002).

### 2.6. Statistical Analysis

The collected data were analyzed using SPSS/WIN 22.0 and AMOS 21.0 programs.

The general characteristics of the subjects and descriptive statistics for each variable were analyzed as descriptive statistics.Correlation between major variables was analyzed using the Pearson Correlation Coefficient.The reliability of the measurement tool was analyzed with Cronbach’s α value.Structural model analysis is a two-step approach, first estimating the measurement model and then measuring the structural model. To verify the validity of the measurement model, confirmatory factor analysis was performed.The maximum likelihood method was used to test the fit of the model. The fitness indices such as χ^2^, TLI, CFI, NFI, IFI, RMSEA, and SRMR were used for the model fit criteria.Bootstrapping was used to verify the significance of the indirect and total effects of the research model.Based on the theoretical basis, a modified model was developed by fixing or liberalizing the path while referring to the parameter estimates and fitness statistics of the hypothetical model, and the final model was presented.

## 3. Results

### 3.1. Patient Characteristics

A total of 209 subjects, with an average age of 61.33 ± 11.21, participated. Seventy were (33.5%) aged 50–59 and 60–69 each; 121 (57.9%) were men; 172 (82.3%) were married; and 155 were unemployed (74.2%), which was the most common. The average elapsed time after surgery was 31.3 ± 27.4 months, with 119 patients (56.9%) at 13–59 months. Of the patients, 79 had cancer in the sigmoid colon (37.8%), and 93 had stage III cancer (44.5%). Also, 140 patients (67%) received chemotherapy, 40 patients (19.1%) received radiotherapy, and 31 patients (14.8%) had a stoma after surgery (Table 1).

### 3.2. Descriptive Statistics and Multicollinearity Test of Measured Variables

Table 2 shows the descriptive statistics of the measurement variables. Each measurement variable satisfies the conditions of a normal distribution with a skewness of −1.19 to 0.83 and kurtosis of −0.93 to 1.17. The Variance Inflation Factor (VIF) value of the measured variable was 1.28–1.92, and the tolerance limit was 0.52–0.78, confirming that there was no problem of multicollinearity.

### 3.3. Validity and Reliability of the Measurement Model 

As a result of confirming the concentration validity of the measurement model, the conceptual reliability showed a value of 0.6 or higher in all variables, and the mean variance extraction was confirmed to be more than 0.5 except for social support and quality of life. The mean variance extraction of social support and quality of life did not meet the standard of 0.5 or more, but it was 0.6 or more in concept reliability, and a standardization coefficient of 0.5 or more was included in the final analysis. In the case of health promotion behavior and quality of life, which had the highest correlation coefficient, the correlation coefficient between the two variables was 0.686, and the square value of the correlation coefficient was 0.471, which was lower than the average variance extraction of the two variables, indicating that there is discriminant validity. In addition, the correlation coefficient between the two variables of health promotion behavior and quality of life was 0.686, with a standard error of 0.037, which was 0.686 ± 2 × 0.037 = −0.049–0.099, so 1 was not included, therefore discriminant validity was satisfied (Table 2). Looking at the correlation based on quality of life, social support (r = 0.48, *p* < 0.001), self-efficacy (r = 0.56, *p* < 0.001), and health promotion behavior (r = 0.69, *p* < 0.001) had a significant net correlation, and the relationship between symptom experience (r = −0.37, *p* < 0.001) showed a significant inverse correlation, securing law validity.

### 3.4. Testing the Hypothetical Model

The fitness index analyzed by the maximum likelihood estimation method of the hypothesis model presented in this study was = 256.936 (df = 95, *p* < 0.001), /df = 2.705, TLI = 0.877, CFI = 0.903, NFI = 0.856, IFI = 0.904, SRMR = 0.076, RMSEA = 0.091 (Figure 2). The fit of the hypothetical model was increased by linking a bidirectional causal relationship to the error term with a high Modification Index (MI). In other words, the measurement model was revised by linking the error terms between health responsibility and exercise, health responsibility and nutrition, health responsibility and rest, and health and nutrition, which are sub-factors of quality of life, and social/family conditions. Also, referring to the modified index, the path from social support to symptom experience and the path from social support to quality of life were deleted. The fit of the modified model is = 181.022 (df = 92, *p* < 0.001), /df = 1.968, TLI = 0.930, CFI = 0.947, NFI = 0.899, IFI = 0.947, SRMR = 0.067, RMSEA = 0.068. The goodness of fit reached the recommended level with some values that did not reach the improved results showed that the modified model was selected as the final model (Figure 3).

### 3.5. The Structural Model

The influencing factor of self-efficacy was social support (γ = 0.448, f = 4.323, *p* < 0.001), and the influencing factor of health promoting behavior was self-efficacy (β = 0.410, f = 7.352 *p* < 0.001) and social support (γ = 0.333, f = 4.053, *p* < 0.001). The factors affecting the quality of life were health promotion behavior (β = 0.386, f = 5.169, *p* < 0.001) and symptom experience (β = −0.051, f = −3.322, *p* < 0.001) was 58.7% (Table 3).

The direct effect, indirect effect, and total effect centered on endogenous variables in the modified model are as shown in (Table 3). As a factor affecting self-efficacy, the direct effect of social support (γ = 0.394, *p* = 0.015) was significant. As the influencing factors of health promotion behavior, the direct effect (γ = 0.344, *p* = 0.004) and the indirect effect (β = 0.191, *p* = 0.004) of social support were both significant, so the total effect (β = 0.535, *p* = 0.004) was significant, and the direct effect (β = 0.481, *p* = 0.011) of self-efficacy showed a significant result.

The factor that directly affected the quality of life, the final variable of this study, was health promotion behavior (β = 0.579, *p* = 0.014), and the direct effect of symptom experience (β = −0.290, *p* = 0.007) was significant, but the indirect effect (β = −0.019, *p* = 0.546) was not significant and the total effect (β = −0.308, *p* = 0.008) showed a significant result. The indirect effect of (β = 0.380, *p* = 0.011) was significant. The direct effect of self-efficacy on quality of life was not significant (β = 0.144, *p* = 0.071), but the indirect effect was significant (β = 0.313, *p* = 0.009) and the total effect was also significant (β = 0.458, *p* = 0.015) (Table 3).

## 4. Discussion

In this study, to develop a structural model for the quality of life of patients undergoing colorectal cancer surgery, a hypothesis model was constructed based on the quality of life model of Wilson and Cleary [4] and a literature review, and a modified model was presented after validating the significance. In the final model, the subject’s social support and symptom experience were set as exogenous variables, and the subject’s self-efficacy, symptom experience, health promotion behavior, and quality of life were set as endogenous variables to verify the fit of the model and the significance of direct and indirect pathways. Therefore, the purpose of this study is to focus on the quality of life of patients undergoing colorectal cancer surgery and the factors that directly affect quality of life.

As a result of this study, it was found that symptom experience and health promotion had a direct effect on the quality of life of colorectal cancer surgery patients, and self-efficacy and social support had an indirect effect, and these variables explained 58.7% of the quality of life of colorectal cancer surgery patients. This is somewhat lower than the explanatory power of the quality of life model, which was 63% to 69%, respectively, in the study results of gastric cancer patients [34] and thyroid cancer patients [10]. It was found that there was a difference according to the influencing factors. The quality of life of the subjects of this study was an average of 91.27 ± 1.30 (average score of 2.69 ± 0.55), which was lower than the 100.23 score of a study [28] that evaluated the quality of life of colorectal cancer patients using the same measurement tool. Among the sub-areas of quality of life, social-family status were the lowest and physical status was the highest, and Kim et al. [28] showed the same results in a study. However, thyroid cancer patients showed the highest result [10] in social-family conditions, indicating that there are differences depending on the type of cancer. The average age of the participants in this study is 61 years, while the average age of 51 years old for thyroid cancer patients is believed to have affected the quality of life. Also, there were differences in the quality of life depending on the presence of a job, the presence of a stoma, and the presence or absence of radiation therapy. In particular, it was reported that the quality of life was lower in the case of radiation therapy or stoma holders [29,35]. Therefore, in order to understand the quality of life of cancer patients in the future, research should be conducted considering the method of controlling variables that affect the selection of subjects and the results to minimize the effect of the distribution of subject characteristics. 

Among the factors affecting quality of life, the direct effect of health promotion behavior (β = 0.588, *p* = 0.009) showed the largest result. In a study [36] of patients with coronary intervention performed based on the health-related quality of life model of Wilson and Cleary [4], health-promoting behavior was reported as a direct influencing factor on quality of life, supporting the results of this study. The health promotion behavior of the subjects of this study had an average score of 2.62 ± 0.61, which was slightly higher than the 2.38 ± 0.32 reported in a study [37] using the same tool for colorectal cancer patients. This difference is interpreted to be a result of the fact that the previous study targeted patients immediately after surgery, and the subjects of this study consisted of patients who completed chemotherapy after surgery and had already adapted to a healthy lifestyle. Health promotion activities such as weight loss, proper physical activity, exercise, and dietary control to maintain health after colon cancer surgery help prevent cancer recurrence and extend survival. The findings reported a 42% reduction in the risk of death in colorectal cancer patients [38]. Therefore, cancer patients should actively practice nutritional management, exercise, and stress management in their daily lives, which is an important nursing intervention that can improve the quality of life of colon cancer survivors. However, in clinical practice, the focus is on the care of colorectal cancer patients who are hospitalized for post-operative care and chemotherapy [39], and when the subject returns to the community after treatment, it is not followed up to monitor whether the patient performs well as a cancer patient. The reality is that you can’t. In the case of colorectal cancer patients who have finished treatment, systematic interventions for nursing care after discharge should be sought so that they can maintain the linkage of health care during the next treatment period.

In terms of self-efficacy, the indirect effect through health promotion behavior was more significant than the direct effect, and the indirect effect through self-efficacy had no direct effect on social support, but significantly affected the quality of life. From these results, it can be seen that cancer patients need support systems such as family or medical support above all else to perform health promotion activities well, and by receiving support, their confidence in overcoming the disease increases. The self-efficacy of the subjects of this study was an average of 45.67 ± 0.64 (mean score of 3.53 ± 0.72), which was an average of 43.39 ± 4.74 (mean score of 4.34 ± 0.47) in a study [40] involving gastric cancer surgery patients using the same tool. The results were similar to those shown. Self-efficacy in cancer patients has been reported as a significant variable directly explaining health promotion behavior [26]. In colorectal cancer patients, self-efficacy was shown to improve quality of life by acting as a mediator of health-related behaviors [19], and exercise programs developed based on self-efficacy theory improved health-related behaviors [41]. In this study, as it acts as a predictive factor for better health promotion behaviors rather than a direct effect on the quality of life in the recovery process of cancer patients, a support system and intervention program to strengthen self-efficacy for cancer patients who need to continue health promotion behaviors are recommended. It should be applied to help maintain and improve quality of life.

In previous studies, social support [11], self-efficacy [13], and health promotion behavior [14] had a causal relationship with symptom experience, which also had an effect on quality of life. Although the path between efficacy and health promotion behavior was established, in this study, these factors had no significant effect, and only the direct effect on quality of life was significant. This can be interpreted as having a greater effect on the quality of life of the symptoms experienced by the subjects than on the indirect effects of the symptom experience on other social and psychological factors. The results of this study were consistent with previous studies [12,27], which reported that the more symptoms experienced, the lower the quality of life. The subjects of this study mainly showed high symptoms such as fatigue, pain, constipation, diarrhea, and distress, which are common symptoms after colorectal cancer surgery [42]. The symptom experience was similar. In light of research results [18] that the more symptoms there are, the more social atrophy and negative emotions are experienced, the lower the quality of life. Also, compared with the results of Baek Young-ae and Lee Myung-sun [43], who used the same tools, the symptoms of the subjects in this study showed a lower level of general symptoms and a higher level of digestive symptoms. This is that the subjects of the previous study were subjects receiving chemotherapy, and 71.9% of subjects in this study completed chemotherapy more than 1 year after surgery.

Considering the above results, it is clear that the health promotion behavior that has the greatest impact on the quality of life of patients after colorectal cancer surgery must be supported by the patient’s confidence that they can do it on their own, and for this, the support system of family and medical personnel is essential.

Symptomatic experience representing the characteristics of colorectal cancer was presented as a factor explaining the quality of life of patients after colorectal cancer surgery, but only had a direct effect on the quality of life. The inability to confirm the mediating effect between the two is a limitation of this study. In previous studies, symptom experience scales including those related to social and psychological symptoms such as depression and anxiety were used. However, since the subjects of this study had completed treatment, the degree of symptom experience was not significant, and although symptom experience directly affected quality of life, but there was no effect on social support or self-efficacy. Although the quality of life requires a time element for change to appear, there may be differences between subjects who are undergoing treatment or who have completed treatment, but symptoms experienced during treatment can significantly decrease over time. In future studies, it is necessary to confirm changes in the treatment process and conduct model validation studies including more social and psychological factors.

## 5. Conclusions

The constructed quality of life model for patients with colorectal cancer surgery was appropriate and had high explanatory power. The pathways that directly affected the quality of life were symptom experience and health promotion, and only indirect effects were confirmed for self-efficacy and social support. These results indicate that quality of life can be improved by lowering the symptoms experienced by patients after colorectal cancer surgery and maintaining health-promoting behaviors. In other words, in order to improve the quality of life of colorectal cancer patients, nursing interventions that can enhance social support and self-efficacy, and nursing interventions that consider influencing factors such as health promotion behaviors are required. We believe that model validation studies that reflect various social and psychological factors are necessary.

In conclusion, in order to improve the quality of life of patients after colon cancer surgery, it is necessary to study specific and viable intervention development that can improve social support and self-efficacy. Considering the variables of symptom experience according to the patient’s recovery stage after colon cancer surgery, follow-up studies on community-based policies and system development are needed to continue health management during the recovery period stage.

## Figures and Tables

**Figure 1 ijerph-19-02564-f001:**
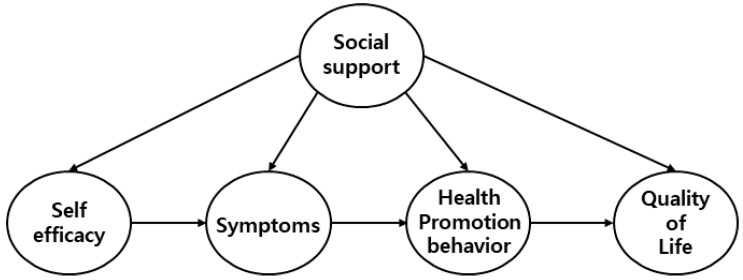
Conceptual framework of this study.

**Figure 2 ijerph-19-02564-f002:**
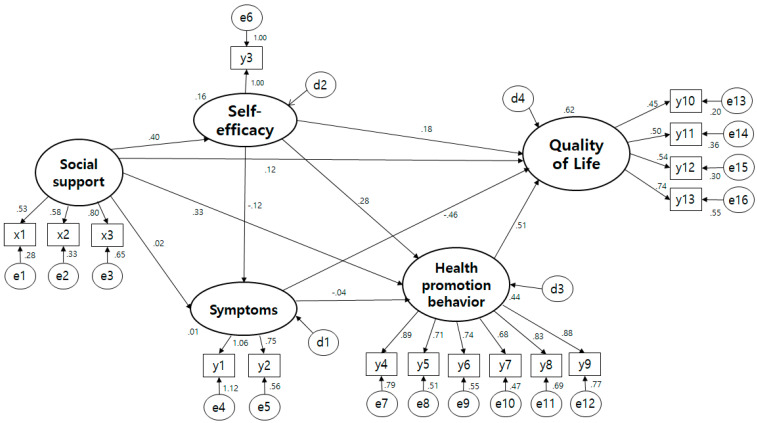
Path diagram hypothetical model. x1 Sinificant support; x2 Family support; x3 Friend support; y1 General symptoms; y2 Gastrointestinal symptoms; y3 Self-efficacy; y4 Self-realization; y5 Health responsibility; y6 Physical activity; y7 Nutrition; y8 Interpersonal relations; y9 Stress management; y10 Physical well being; y11 Social/Family well being; y12 Emotional well being; y13 Functional well being.

**Figure 3 ijerph-19-02564-f003:**
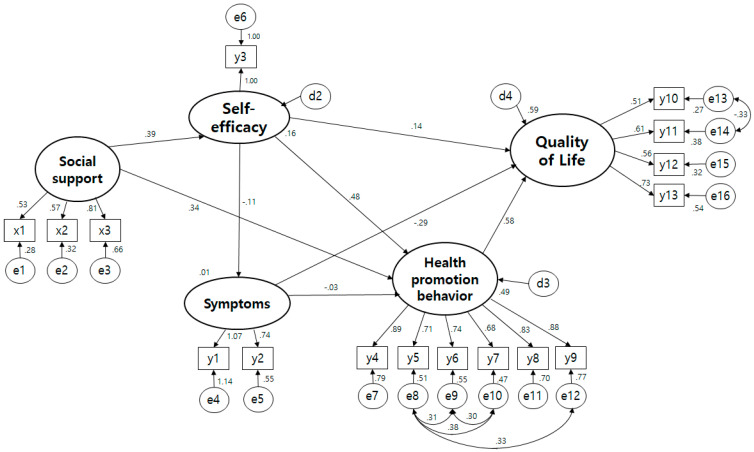
Path diagram modified model. x1 Sinificant support; x2 Family support; x3 Friend support; y1 General symptoms; y2 Gastrointestinal symptoms; y3 Self-efficacy; y4 Self-realization; y5 Health responsibility; y6 Physical activity; y7 Nutrition; y8 Interpersonal relations; y9 Stress management; y10 Physical well being; y11 Social/Family well being; y12 Emotional well being; y13 Functional well being.

**Table 1 ijerph-19-02564-t001:** Characteristics of Subjects and Research Concept Variables.

Characteristics	Categories	*n* (%)	M ± SD
Age (year)	≥49	23 (11.0)	61.33 ± 11.21
	50~59	70 (33.5)	
	60~69	70 (33.5)	
	≤70	46 (22.0)	
Gender	Male	121 (57.9)	
	Female	88 (42.1)	
Marriage status	Unmarried	11 (5.3)	
	Married	172 (82.3)	
	Diverse/Separation	26 (12.4)	
Occupation	Yes	54 (25.8)	
	No	155 (74.2)	
Period of received Surgery	≥5	34 (16.3)	31.3 ± 27.4
(month)	6–12	31 (14.8)	
	13–59	119 (56.9)	
	≤60	25 (12.0)	
Stage	I	36 (17.2)	
	II	53 (25.4)	
	III	93 (44.5)	
	IV	27 (12.9)	
Cancer location	Ascending colon	56 (26.8)	
	Descending colon	12 (5.7)	
	Sigmoid colon	79 (37.8)	
	Rectum	62 (29.7)	
Stoma	Yes	31 (14.8)	
	No	178 (85.2)	
Chemotherapy	Yes	140 (67.0)	
	No	69 (33.0)	
Radiation therapy	Yes	40 (19.1)	
	No	169 (80.9)	

SD = standard deviation.

**Table 2 ijerph-19-02564-t002:** Descriptive Statistics of Variables.

Variable	Categories	M ± SD	Skewmess	Kurtosis	Tolerance	VIF	Estimate	CR	AVE
Social support	Significant	2.53 ± 1.19	1.40	−0.93			0.529		
	Family	4.04 ± 0.90	−1.00	0.76			0.577		
	Friend	3.30 ± 1.04	−0.22	−0.63			0.804		
	Total	39.45 ± 0.67	0.12	−0.17	0.78	1.28		0.677	0.420
Symptoms	General	2.61 ± 2.16	0.83	−0.30			1.031		
	G-I	2.57 ± 2.31	0.80	−0.34			0.797		
	Total	46.84 ± 7.81	0.80	−0.34	0.59	1.68		0.865	0.686
Self-efficacy	Total	45.67 ± 0.64	−0.70	0.45	0.63	1.59	0.312	0.661	0.661
Health Promotion behavior	Self-realization	2.60 ± 0.69	−0.10	−0.36			0.892		
	Heath responsibility	2.38 ± 0.60	0.15	−0.45			0.715		
	Physical activity	2.42 ± 0.86	0.12	−0.94			0.743		
	Nutrition	2.67 ± 0.66	−0.02	−0.52			0.683		
	Interpersonal relations	2.69 ± 0.68	−0.23	−0.23			0.833		
	Stress management	2.92 ± 0.80	0.14	−0.45			0.880		
	Total	83.68 ± 1.33	0.03	−0.33	0.55	1.81		0.919	0.657
Quality of Life	Physical well being	3.13 ± 0.79	−1.19	1.17			0.525		
	Social/Family well being	2.30 ± 0.85	−0.50	−0.31			0.639		
	Emotional well being	2.93 ± 0.69	−0.55	−0.16			0.561		
	Functional well being	2.55 ± 0.97	−0.40	−0.49			0.720		
	Total	73.35 ± 1.30	−0.12	−0.63	0.52	1.92		0.785	0.480

SD = standard deviation; VIF = variance inflation factor; CR = critical ratio; AVE = average variance extracted.

**Table 3 ijerph-19-02564-t003:** Modified model of Standardized Regression Weights.

Pathway			SE	T(*p*)	SMC	Direct Effect (*p*)	Indirect Effect (*p*)	Total Effect (*p*)
Self-efficacy	←	Social support	0.448	4.323 (<0.001)	0.155	0.394 (0.015)		0.394 (0.015)
Symptoms	←	Social support					−0.045 (0.093)	−0.045 (0.093)
	←	Self-efficacy	−0.363	−1.773 (0.076)	0.130	−0.113 (0.104)		−0.113 (0.104)
Health promotion behavior	←	Social support	0.333	4.053 (<0.001)	0.486	0.344 (0.004)	0.191 (0.004)	0.535 (0.004)
	←	Symptoms	−0.009	−1.625 (0.532)		−0.032 (0.528)		−0.032 (0.528)
	←	Self-efficacy	0.410	7.352 (<0.001)		0.481 (0.011)		0.481 (0.011)
Quality of Life	←	Social support			0.587		0.380 (0.011)	0.380 (0.011)
	←	Health Promotion behavior	0.386	5.169 (<0.001)		0.579 (0.014)		0.579 (0.014)
	←	Self-efficacy	0.82	1.773 (0.076)		0.144 (0.071)	0.313 (0.009)	0.458 (0.015)
	←	Symptoms	−0.051	−3.322 (<0.001)		−0.290 (0.007)	−0.019 (0.546)	−0.308 (0.008)

SE = standard estimates; SMC = squared multiple correlation.

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
