# Peer review of "A Structural Model of Quality of Life in Patients after Colorectal Cancer Surgery"

_ijerph, 2022, doi:10.3390/ijerph19052564_

Round 1

Reviewer 1 Report

This study investigated a hypothetical model for the factors 10 affecting the quality of life of postoperative colorectal cancer patients. They found that strengthening patients’ health promotion behaviors to alleviate their symptoms and improving their social support and self-efficacy would have benefits to improve the quality of life of colorectal cancer patients.

This manuscript is well written and described. However, there are some minor errors needed to be revised:

1, line56, missing preposition.

2, line149-166, repetitive paragraphs.

3, the resolution of Figures 1 and 2 is too low.

Author Response

비싼. 의사.

나는 당신의 조언을 깊이 감사드립니다.
당신은 내 종이를 더 잘 만들었습니다.
정말 고마워요.

1, line56, 누락 된 전치소. - 두 문장을 결합.

2, line149-166, 반복단락. - 중복을 제거합니다.

도 3, 그림 1과 2의 해상도가 너무 낮습니다. - 더 나은 해상도를 갖도록 그림을 변경했습니다.

Reviewer 2 Report

This is an interesting report and the article is well written. However, in order to arouse the readers' interest whether interest to the readers of IJERPH, the authors are adviced to clarify even more:

  • Please clarify why the authors have chosen a single model by Wilson and Cleary. Are there any comparisons to others?
  • There no dout that the work is novel, but please explain for the readers why the number of 209 patients analyzed sufficient for the work? Many differences in the patients could account for the observed findings. To make these data meaningful, please add in the premeasuring of patient’s demographics more information regarding systemic diseases. Please if these data not availabe, then add this as limitations.
  • The most important, please add to the path diagrams complete legends and for the clarification detailed information (figures). 

Author Response

Dear. Doctor.

I deeply appreciate your advice.
You made my paper better.
Thank you so much.

1. Please clarify why the authors have chosen a single model by Wilson and Cleary. Are there any comparisons to others?

-This model was developed by the need for planning and intervention to improve the patient's quality of life. In order to further understand the effect on the quality of life, it can be said to be useful for explaining the quality of life of colorectal cancer patients as it evaluates the overall quality of life based on diseases and includes symptoms that show the unique characteristics of the disease.

2. There no dout that the work is novel, but please explain for the readers why the number of 209 patients analyzed sufficient for the work? Many differences in the patients could account for the observed findings. To make these data meaningful, please add in the premeasuring of patient’s demographics more information regarding systemic diseases. Please if these data not availabe, then add this as limitations.

-Based on Kline's evidence, the sample size of the SEM was calculated because more than 200 was desirable.

-Since the patient's age, stage, and physiological characteristics are unique data that cannot be mediated, we judged that it was not right to choose as variables.

3. The most important, please add to the path diagrams complete legends and for the clarification detailed information (figures).                                 

- I added a conceptual path and detailed information.

Reviewer 3 Report

Review for the Manuscript A Structural Model of Quality of Life in Patients after Colorectal Cancer

Dear Doctor, thank you very much for the opportunity to review this manuscript.

General comments:

This is an interesting manuscript that is concerned with Colorectal cancer, which is an important issue in modern societies worldwide. Although many things are interesting in this manuscript, I have some suggestions for the authors.

Abstract

            I suggest the authors improve this section since I do not feel that it brings enough information to the readers to understand entirely the findings of the manuscript.

Introduction

  • In the first paragraph, the authors say that “Colorectal cancer is the second most common cancer among men and women in Ko- 24 rea as lifestyles become more westernized, and the incidence rate is increasing every year 25 [1].”.

I agree with this sentence. Nevertheless, I suggest including more references and if it is possible, include references from 2021-2022 (if it is possible to find). Please, check in Pubmed if there are new epidemiological studies on colorectal cancer.

  • In the second paragraph, we can see: “Quality of life refers to a subjective sense of well-being for life [4]. In particular, in order to evaluate and improve the quality of life of cancer patients, it is a factor that affects the quality of life. It has been evaluated by including factors and the social context to which the patient belongs [5]. Recently, the importance of quality of life as an indicator or predictor of survival of cancer patients has been emphasized [6], and the improvement of health promotion behaviors and quality of life of the subject is being evaluated in relation to it [7].”

            I suggest that the authors re-write this paragraph, mainly in this part: “Quality of life refers to a subjective sense of well-being for life [4]. In particular, in order to evaluate and improve the quality of life of cancer patients, it is a factor that affects the quality of life…”. Please, re-write to facilitate the understanding of these sentences.

  • In the second page, the first paragraph we can read: “As for factors affecting the quality of life of colorectal cancer patients, social support 54 [11], symptom experience [12], self-efficacy [13], and health promotion behavior [14] were 55 reported as major variables. Wilson and Cleary’s model [4]...”

            Please see this last sentence: Wilson and Cleary’s model [4]. There is information missing here. Please, check and improve.

            In the sentence: “…patients in Korea. In Korea, the number of colorectal cancer patients has recently increased, and the 5-year survival rate has increased compared to other cancers [1].”

            I suggest: “…patients in Korea. In this country, the number of colorectal cancer patients has recently increased…”

  • There are terms repeated many times along the Introduction, such as “quality of life”. Please, check.

  • Please, see this sentence: “Among the factors suggested by Wilson and Cleary [4], factors influencing the quality of life of colorectal cancer surgery patients were selected as relating to self-efficacy, and environmental characteristic variables were selected as relating to social support. Symptom state variables were selected as related to 106 symptom experience, and functional state variables were selected as related to health pro-motion behavior, and a hypothetical model of this study was established. According to the theory of Wilson and Cleary [4]...”

Since the authors used reference 4 for both sentences, please, cite Wilson and Cleary only once.

  • I also suggest reducing the Introduction section. There are several information in this section that could be in the Discussion section.

  • Figure 1 is not in good quality to be published in this format.

AIMS

  • Research aims

“This study aims to investigate the factors affecting the quality of life of patients after colorectal cancer surgery based on the health-related quality of life model of Wilson and  Cleary [4]. The specific purpose of this study is as follows: First, we present a hypothetical model for the quality of life of colorectal cancer surgery patients. Second, we present a modified model that verifies the suitability between the hypothetical model and the actual data and better explains the relationship between related variables. Third, the direct and indirect effects of variables affecting the quality of life of colorectal cancer patients are identified”

I suggest living in this section only the sentence: “This study aims to investigate the factors affecting the quality of life of patients after colorectal cancer surgery based on the health-related quality of life model of Wilson and  Cleary…”

This part: “The specific purpose of this study is as follows: First, we present a hypothetical model for the quality of life of colorectal cancer sur gery patients. Second, we present a modified model that verifies the suitability between the hypothetical model and the actual data and better explains the relationship between related variables. Third, the direct and indirect effects of variables affecting the quality of life of colorectal cancer patients are identified” should be in the Methods section.

  • Materials and Methods

In this section we see:

“2.1. Research design

This study is based on the health-related quality of life model of Wilson and Cleary [4] and previous studies”.

I suggest including references for these previous studies.

In the criteria for selection: We see “visited S University Hospital in Gyeonggi-do”… Is it corrected to say “S University”?

  1. Results

In table 1: please, include a space before each parenthesis.

Figure 2 should also be improved in the quality of the image.

  1. Conclusion

The sentence “This study is a structural model study to verify the hypothetical model constructed for the quality of life of colorectal cancer surgery patients” should be removed from this section.

The sentence “Considering the variables of symptom experience according to the patient's recovery stage after colon cancer surgery, follow-up studies on community-based policies and system development are proposed to continue health management during the recovery period stage”

Did the authors mean that “Considering the variables of symptom experience according to the patient's recovery stage after colon cancer surgery, follow-up studies on community-based policies and system development are needed to continue health management during the recovery period stage”?

Author Response

Dear. Doctor.

I deeply appreciate your advice.
You made my paper better.
Thank you so much.

1. Abstract : I suggest the authors improve this section since I do not feel that it brings enough information to the readers to understand entirely the findings of the manuscript.

-The result part was further modified.

2. Introduction

In the first paragraph, the authors say that “Colorectal cancer is the second most common cancer among men and women in Ko- 24 rea as lifestyles become more westernized, and the incidence rate is increasing every year 25 [1].”.

I agree with this sentence. Nevertheless, I suggest including more references and if it is possible, include references from 2021-2022 (if it is possible to find). Please, check in Pubmed if there are new epidemiological studies on colorectal cancer.

-I revised the contents as you advised.

3. In the second paragraph, we can see: “Quality of life refers to a subjective sense of well-being for life [4]. In particular, in order to evaluate and improve the quality of life of cancer patients, it is a factor that affects the quality of life. It has been evaluated by including factors and the social context to which the patient belongs [5]. Recently, the importance of quality of life as an indicator or predictor of survival of cancer patients has been emphasized [6], and the improvement of health promotion behaviors and quality of life of the subject is being evaluated in relation to it [7].”

I suggest that the authors re-write this paragraph, mainly in this part: “Quality of life refers to a subjective sense of well-being for life [4]. In particular, in order to evaluate and improve the quality of life of cancer patients, it is a factor that affects the quality of life…”. Please, rewrite to facilitate the understanding of these sentences.

Quality of life means a subjective sense of well-being in one's life [4]. In particular, in order to evaluate and improve the quality of life of cancer patients, it has been evaluated including various factors such as physical, psychological, social well-being, and symptoms, as well as the social context in which the patient belongs [5].

-I revised the contents as you advised.

4.In the second page, the first paragraph we can read: “As for factors affecting the quality of life of colorectal cancer patients, social support 54 [11], symptom experience [12], self-efficacy [13], and health promotion behavior [14] were 55 reported as major variables. Wilson and Cleary’s model [4]...”

Please see this last sentence: Wilson and Cleary’s model [4]. There is information missing here. Please, check and improve.

-I revised the contents as you advised.

5. In the sentence: “…patients in Korea. In Korea, the number of colorectal cancer patients has recently increased, and the 5-year survival rate has increased compared to other cancers [1].”

I suggest: “…patients in Korea. In this country, the number of colorectal cancer patients has recently increased…”

-I revised the contents as you advised.

6. There are terms repeated many times along the Introduction, such as “quality of life”. Please, check.

Since the authors used reference 4 for both sentences, please, cite Wilson and Cleary only once.

-I revised the contents as you advised.

7. Figure 1 is not in good quality to be published in this format.

- I changed the figures to have better resolution

8. I suggest living in this section only the sentence: “This study aims to investigate the factors affecting the quality of life of patients after colorectal cancer surgery based on the health-related quality of life model of Wilson and Cleary…”

This part: “The specific purpose of this study is as follows: First, we present a hypothetical model for the quality of life of colorectal cancer sur gery patients. Second, we present a modified model that verifies the suitability between the hypothetical model and the actual data and better explains the relationship between related variables. Third, the direct and indirect effects of variables affecting the quality of life of colorectal cancer patients are identified” should be in the Methods section.

-I added it in as a method section.

9. This study is based on the health-related quality of life model of Wilson and Cleary [4] and previous studies”.

I suggest including references for these previous studies.

-I added a reference as you advised.

10. In the criteria for selection: We see “visited S University Hospital in Gyeonggi-do”… Is it corrected to say “S University”?

- The expression of Soonchunhyeon University Hospital is correct.

11. Results

In table 1: please, include a space before each parenthesis.

-As you advised, I included a space in front of each parenthesis.

12. Figure 2 should also be improved in the quality of the image.

- I changed the figures to have better resolution

13.Conclusion

The sentence “This study is a structural model study to verify the hypothetical model constructed for the quality of life of colorectal cancer surgery patients” should be removed from this section.

-I deleted the sentence as you advised.

14. The sentence “Considering the variables of symptom experience according to the patient's recovery stage after colon cancer surgery, follow-up studies on community-based policies and system development are proposed to continue health management during the recovery period stage”

Did the authors mean that “Considering the variables of symptom experience according to the patient's recovery stage after colon cancer surgery, follow-up studies on community-based policies and system development are needed to continue health management during the recovery period stage”?

-Yes, your advice is correct. I revised it as your advice.

Round 2

Reviewer 2 Report

Accept.

Reviewer 3 Report

Dear doctor,
I recommend the manuscript for publication.